# OpenReview forum: "ShapeUQ: Propagating 3D Reconstruction Uncertainty Through Scientific PDE Simulations via Shape Calculus"
_thecvf.com/CVPR/2026/Workshop/3D4S — CVPR 2026 Workshop 3D4S Oral_

### Official Review · Reviewer_3MwQ · 2026-04-22
**strong idea but further studies are needed**

**Rating:** 8
**Confidence:** 3

**Review:**

This paper investigates how uncertainty in SDF 3D reconstruction is propagated to downstream scientific PDE simulations. It proposes the SHAPEUQ framework, which converts reconstruction uncertainty into uncertainty in downstream simulations at a lower computational cost, offering significant research and practical value.
The main limitation of this paper is that the estimation using GP is overly simplified, neglecting non-Gaussian boundaries (such as holes and sharp corners) found in real-world problems; these geometries were not tested in the experiments. Furthermore, first-order PDEs in real-world applications have significant limitations, whereas nonlinear PDEs are more challenging and hold greater practical value.

---

### Official Review · Reviewer_RMiB · 2026-04-23
**Review of ShapeUQ**

**Rating:** 6
**Confidence:** 2

**Review:**

### Summary
The paper proposes a shape-calculus-based method for propagating neural-SDF reconstruction uncertainty into downstream PDE solution uncertainty
### Strengths
- The use of Hadamard shape derivatives and an adjoint solve is computationally attractive. The uncertainty propagation can be done with one additional PDE solve rather than many Monte Carlo geometry samples.
- The experimental section is also relatively convincing. The paper evaluates glacier, protein, and coral geometries, and reports close-to-nominal 90% coverage with 14×–31× speedups over MC-500.


### Weaknesses
- Reconstruction errors are rarely smoothly distributed Gaussian noise. They manifest as highly systematic artifacts, topological floaters, or missing structures due to occlusions. It is unclear if the first-order Taylor expansion will hold under these non-Gaussian, topologically disruptive errors, especially when local noise exceeds the curvature of the boundary.
- The PDE setting is also still relatively restricted. Much of the theory is clearest for linear elliptic problems, and extension to nonlinear PDEs is left for future work.

---

### Decision · Program_Chairs · 2026-04-28

Accept (Oral)